# Carvedilol targets β-arrestins to rewire innate immunity and improve oncolytic adenoviral therapy

Joseph I. Hoare[1], Bleona Osmani[1], Emily A. O'Sullivan[1], Ashley Browne[1], Nicola Campbell[1], Stephen Metcalf[1], Francesco Nicolini[1], Jayeta Saxena[1], Sarah A. Martin [1] & Michelle Lockley [1,2✉]

Oncolytic viruses are being tested in clinical trials, including in women with ovarian cancer. We use a drug-repurposing approach to identify existing drugs that enhance the activity of oncolytic adenoviruses. This reveals that carvedilol, a β-arrestin-biased β-blocker, synergises with both wild-type adenovirus and the E1A-CR2-deleted oncolytic adenovirus, *dl*922-947. Synergy is not due to β-adrenergic blockade but is dependent on β-arrestins and is reversed by β-arrestin CRISPR gene editing. Co-treatment with *dl*922-947 and carvedilol causes increased viral DNA replication, greater viral protein expression and higher titres of infectious viral particles. Carvedilol also enhances viral efficacy in orthotopic, intraperitoneal murine models, achieving more rapid tumour clearance than virus alone. Increased anti-cancer activity is associated with an intratumoural inflammatory cell infiltrate and systemic cytokine release. In summary, carvedilol augments the activity of oncolytic adenoviruses via β-arrestins to re-wire cytokine networks and innate immunity and could therefore improve oncolytic viruses for cancer patient treatment.

[1] Centre for Cancer Cell and Molecular Biology, Barts Cancer Institute, Queen Mary University of London, London, UK. [2] Department of Gynaecological Oncology, Cancer Services, University College London Hospital, London, UK. ✉email: m.lockley@qmul.ac.uk

Ovarian cancer is the 7th most common cancer in women worldwide[1]. The most frequent histopathological subtype is high-grade serous cancer (HGSC), accounting for 70% of cases[2]. Platinum-based chemotherapy is used repeatedly throughout the disease course but patients respond less well to successive treatment and 85% ultimately die with platinum-resistant cancer[3]. As with many solid cancers, there is an urgent and unmet clinical need for new therapeutic approaches.

Oncolytic viruses are a promising new anti-cancer treatment. These viruses replicate selectively within cells with matched genetic defects causing cell death and dissemination of virions to neighbouring cells[4,5]. In addition to direct cell lysis, oncolytic viruses trigger an anti-cancer immune response[6,7] that can be tailored by genetically modifying and arming viruses[8,9] or through therapeutic combination with other immunomodulatory agents[10–12]. Currently, several viruses are undergoing evaluation in clinical trials[13] including in ovarian cancer[14–17]. Moreover, the clinical utility of oncolytic viruses in cancer therapy has been endorsed by FDA approval of the modified herpes virus, T-VEC, for the treatment of metastatic malignant melanoma[18].

The oncolytic adenovirus, *dl*922-947 is a type 5 adenovirus with deletion of the E1A-CR2 genomic region[19]. This region initiates viral replication by activating the cellular pRb/G1-S cell cycle checkpoint and so *dl*922-947 is proposed to replicate selectively in cells with Rb checkpoint abnormalities, such as malignant cells[20]. We have previously shown that *dl*922-947 has activity in ovarian cancer[21–23] including in chemotherapy-resistant disease[24]. In order to facilitate clinical implementation of *dl*922-947 and other oncolytic viruses, we utilised a compound library screen to identify existing drugs that could enhance oncolytic adenoviral activity. Using this approach, we identified the β-adrenergic receptor (β-AR) antagonist, carvedilol.

The β1 and β2-ARs are seven transmembrane G protein-coupled receptors that are activated by catecholamine binding and selectively couple to G protein complexes that are classified based on their α-subunits into: Gα-stimulatory (Gαs), Gα-inhibitory (Gαi), Gαq/11, and Gα12/135[25]. Phosphorylation of activated β-ARs by G protein-coupled receptor kinases (GRKs) leads to recruitment of the regulatory proteins, β-arrestin 1 and β-arrestin 2, which control adrenergic signalling by terminating G protein activation. β-arrestins also function as essential scaffold proteins, regulating diverse intracellular signalling networks[26] such as mitogen-activated protein kinases (MAPK) cascades including ERK1/2[27,28], as well as phosphoinositol kinase (PI3K) and Akt[26]. Moreover, β-arrestins have also been shown to modulate inflammation via NFκB[29,30].

β-blocking drugs interact with β1-AR and β2-AR and thus influence downstream signalling via both G proteins and β-arrestins. By blocking the effects of excess catecholamine stimulation these drugs have found widespread use in the treatment of a variety of cardiovascular diseases. Interestingly, epidemiological studies have shown that β-blocker use correlates with improved ovarian cancer patient survival[31,32] although the precise underlying mechanisms have not been defined and the specific effect of carvedilol is unknown. Carvedilol is a non-selective inhibitor of β1-AR, β2-AR and also the α1-AR. Unusually, and in marked contrast to other β-blockers, carvedilol antagonises β-adrenergic signalling via G proteins, while simultaneously promoting recruitment of Gαi specifically to β1 and not β2-ARs, to initiate β-arrestin-mediated downstream signalling pathways. This uncommon mode of action is referred to as β-arrestin-biased β-adrenergic antagonism[33].

Here we show that carvedilol synergises with oncolytic adenoviruses via β-arrestins to rewire inflammatory signalling networks. Our work demonstrates that β-arrestin-targeted therapies such as carvedilol can significantly improve the activity of oncolytic adenoviruses. Our discovery has clear implications for optimisation and clinical implementation of oncolytic viral therapies.

## Results

**Carvedilol synergises with adenoviruses in HGSC.** We screened a library of 1177 compounds to identify drugs that can synergise with *dl*922-947 in HGSC. Two drug screen experiments were conducted, the first comparing OVCAR4 with cisplatin-resistant Ov4Cis cells and the second comparing OVCAR4 with carboplatin-resistant Ov4Carbo cells. Cell viability was assessed using CellTiter-Glo® and the effect of each compound was determined by comparing luminescence output following drug alone, *dl*922-947, drug + *dl*922-947 and vehicle. This effect was represented as a $\log_2$-surviving fraction (s.f.) and expressed relative to the median s.f. for the entire 96-well plate (Supplementary Data 1 and 2). Potential 'hit' compounds were identified that fulfilled our pre-defined criteria of s.f. $< -2$ following drug + *dl*922-947 together with s.f. $> -2$ for drug alone. Using these criteria, six drugs were identified as potential 'hits' in both Ov4Cis and Ov4Carbo cells (Supplementary Data 3). Out of these six drugs, carvedilol was selected for further investigation because it is an oral agent with a well-described mechanism of action and a favourable side effect profile.

Carvedilol was validated in multiple repeat experiments in which it enhanced the effect of *dl*922-947 (Fig. 1a and Supplementary Fig. 1a–c), significantly reducing cell survival and viral $IC_{50}$ compared to *dl*922-947 alone: platinum-sensitive OVCAR4 cells (2.9-fold reduction; $P = 0.0185$), cisplatin-resistant Ov4Cis (7.9-fold; $P = 0.0082$) and carboplatin-resistant Ov4Carbo (7.4-fold; $P = 0.0439$). Carvedilol also improved the efficacy of *dl*922-947 in both cisplatin-sensitive Cov318 (4.1-fold; $P = 0.0408$) and cisplatin-resistant CovCis cells (3.9-fold; $P = 0.1597$) (Fig. 1b and Supplementary Fig. 1d, e). We also tested carvedilol in combination with wild-type human adenovirus type 5 (AD5-WT) and once again found that $IC_{50}$ was reduced in Ov4Cis cells (38.7-fold; $P = 0.0625$) and Ov4Carbo (14.4-fold; $P = 0.0625$) (Supplementary Fig. 2).

To determine whether this combination was additive or synergistic, we performed Combination Index (CI) analysis using CompuSyn software where a CI $> 1$ indicates synergy, CI $= 1$ indicates addition, and a CI $< 1$ indicates antagonism[34]. Chou–Talalay isobolograms confirmed that carvedilol (3 μM and 10 μM) and *dl*922-947 were synergistic at viral doses greater than MOI 1 in sensitive OVCAR4 cells. Synergy was even more marked in the two platinum-resistant cell lines, occurring at viral MOI above 0.1 (Fig. 1c). Importantly, carvedilol also synergised with *dl*922-947 in Cov318 and CovCis cells (viral doses > MOI 1) (Fig. 1h) and was also able to synergise with AD5-WT in the OVCAR4-derived cell panel (viral doses >10 MOI) (Supplementary Fig. 2).

Colony formation assays again showed that 1, 3 and 10 μM carvedilol reduced the $IC_{50}$ of *dl*922-947 in all five cell lines tested: OVCAR4 (27.3-fold, $P = 0.0058$), Ov4Cis (11.1-fold, $P = 0.0004$), Ov4Carbo (55.2-fold, $P = 0.0007$), Cov318 (2.7-fold, $P = 0.0625$), and CovCis (63.2-fold, $P = 0.0313$) (Supplementary Fig. 3). These data therefore strongly endorse carvedilol as a potential synergistic adjuvant to enhance oncolytic adenoviral efficacy in HGSC, including in platinum-resistant disease.

**Carvedilol increases virus-induced cell death by promoting viral life cycle.** To understand the mechanism by which carvedilol synergised with *dl*922-947, we interrogated carboplatin-resistant Ov4Carbo cells because this cell line displayed the greatest synergistic effect and is representative of the more challenging clinical scenario of platinum-resistant HGSC. Neither

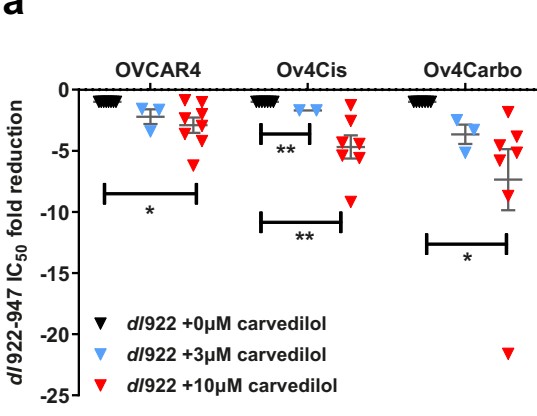

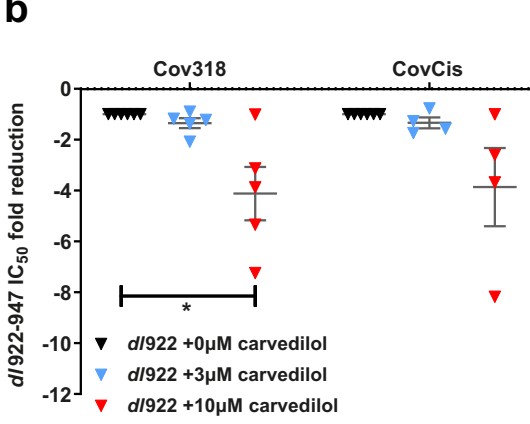

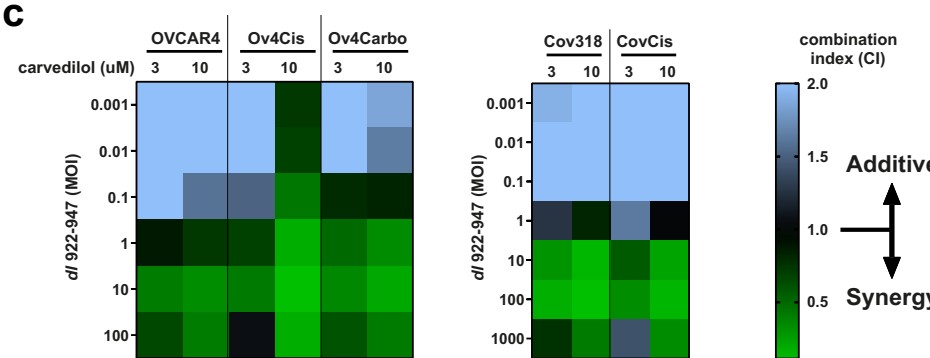

**Fig. 1 Carvedilol synergises with *dl*922-947 in HGSC cell lines. a** Fold reductions of *dl*922-947 IC$_{50}$ in CTG assays following carvedilol treatment compared to virus alone in the OVCAR4 panel (OVCAR4 +vehicle: $n = 8$, +3 μM: $n = 3$, +10 μM: $n = 8$; Ov4Cis +vehicle: $n = 7$, +3 μM: $n = 3$, +10 μM: $n = 7$; Ov4Carbo +vehicle: $n = 8$, +3 μM: $n = 3$, +10 μM: $n = 8$ biological repeats; mean ± SEM; paired *t*-test, *$P \leq 0.05$; **$P \leq 0.01$). **b** Fold reductions of *dl*922-947 IC$_{50}$ in CTG assays following carvedilol treatment compared to virus alone in Cov318 panel (Cov318: $n = 5$; CovCis: $n = 4$ biological repeat; mean ± SEM; paired *t*-test, *$P \leq 0.05$). **c** Combination Index heatmaps between carvedilol and *dl*922-947 calculated from ATP-based viability assays.

viral attachment, measured by quantitative PCR (qPCR) for viral hexon DNA after 1 h at 4 °C (Fig. 2a), or viral entry, measured by quantifying green fluorescent viral particles with FACS at 24 h (Fig. 2b), were affected by the addition of carvedilol.

In marked contrast, western blot for adenoviral E1A and structural proteins revealed that carvedilol significantly increased expression of these viral proteins 96 and 120 h post infection (Fig. 2c and Supplementary Fig. 4a). Densitometry quantification confirmed significant increases in E1A (96 h: $P = 0.0168$; 120 h: $P = 0.04$) (Fig. 2d) and structural viral proteins (96 h: $P = 0.0147$; 120 h: $P = 0.0498$) (Fig. 2e) following carvedilol co-treatment. We also measured viral DNA replication by qPCR for viral hexon DNA. In keeping with the increase in viral protein expression, carvedilol significantly enhanced viral DNA replication at both 120 ($P = 0.0422$), and 144 h post infection (hpi) ($P = 0.0213$) compared to virus treatment alone (Fig. 2f).

Next, to investigate the influence of carvedilol on production of complete infectious virions, we treated Ov4Carbo cells with *dl*922-947 either alone or with carvedilol and quantified viral titre (pfu/ml) by limiting dilution assays. Carvedilol-treated samples consistently displayed higher pfu/ml compared to virus alone in three biological repeat experiments with average fold increases of 3.97-fold at 120 hpi (Fig. 2g) and 3.90-fold at 144 hpi (Fig. 2h). Together, these data demonstrate that carvedilol enhances viral efficacy by promoting viral replication within ovarian cancer cells.

**Synergy between *dl*922-947 and carvedilol is mediated by β-arrestins**. Having determined that carvedilol synergises with adenovirus by promoting viral replication, we next investigated the underlying biological mechanisms. A review of our drug library identified a further twelve drugs in our screen with known activity as β-adrenergic antagonists as well as nine with potential activity as α-adrenergic antagonists. None of these agents met the criteria we predefined to identify potential hit drugs. To further investigate whether β-adrenergic activity was important for viral synergy we tested the non-selective β-blocker propranolol and the β-adrenergic receptor agonist isoprenaline in cell viability validation experiments using the same protocol as the initial drug screen. Interestingly, we found that neither of these drugs significantly altered viral efficacy (Supplementary Fig. 5).

Carvedilol is unusual in that it exhibits β-arrestin-biased agonism. A literature search revealed only one other β-blocker, alprenolol that shares this signalling mechanism[35]. Prior to investigating whether alprenolol could reproduce the synergy we had observed with carvedilol, we first tested the sensitivity of HGSC cell lines to alprenolol alone. These experiments revealed that IC$_{50}$ to alprenolol was ~10-fold higher than to carvedilol in all three cell lines tested (Supplementary Fig. 6a) therefore 50 μM and 100 μm alprenolol were used for combination cell viability assays in vitro. These experiments confirmed that, like carvedilol, alprenolol also synergised with *dl*922-947. Alprenolol

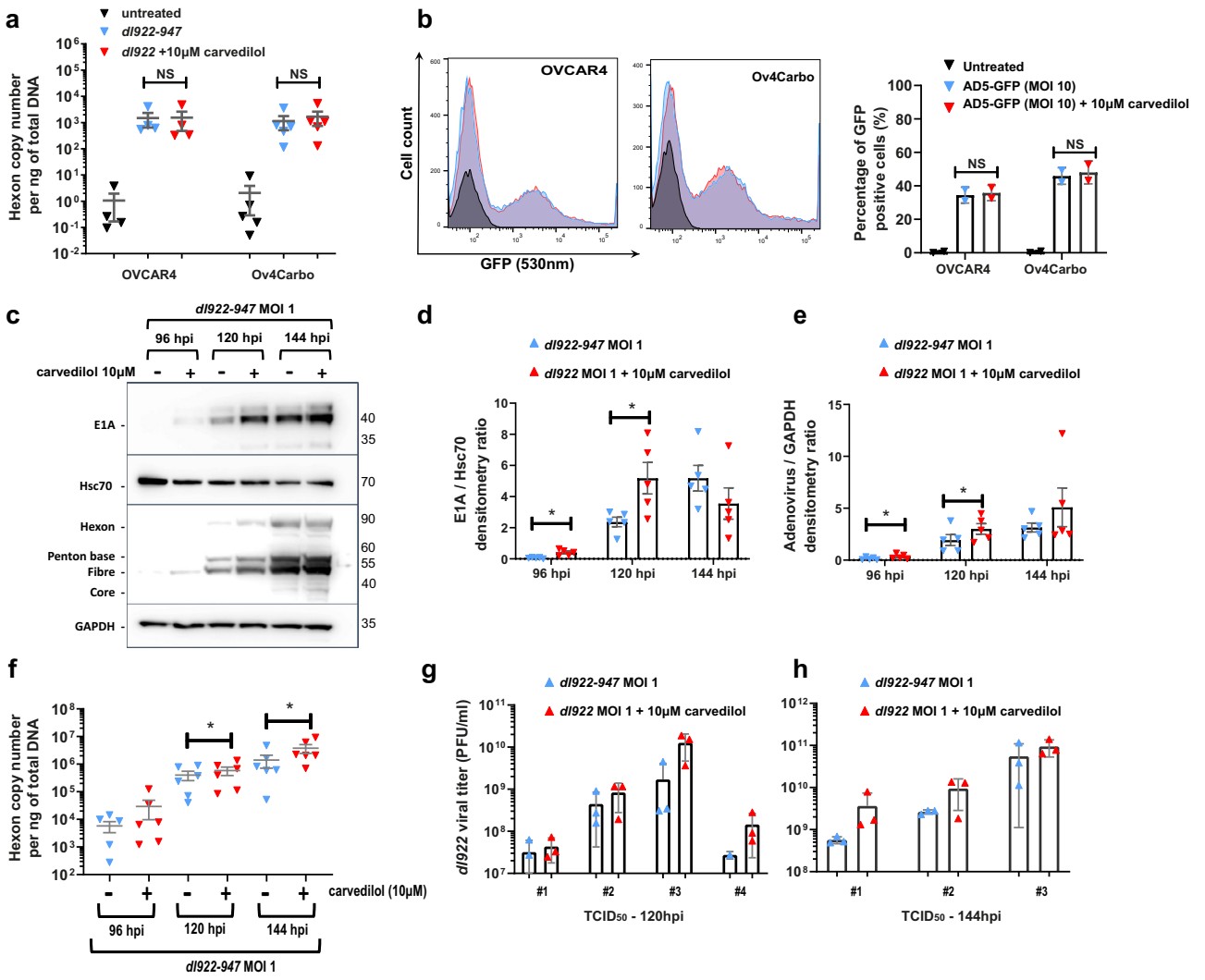

**Fig. 2 Carvedilol enhances viral DNA and protein expression in Ov4Carbo cells. a** Viral attachment 1 h post infection (hpi) measured by qPCR for viral Hexon over time following carvedilol (10 µM) treatment in OVCAR4 ($n = 4$ biological repeats) and Ov4carbo ($n = 5$ biological repeats) cells (ns $P > 0.05$). **b** Flow cytometry analysis of viral infectivity. Percentages of GFP positive OVCAR4 and Ov4Carbo cell populations measured 24 h after treatment with AD5-GFP (MOI 10) and carvedilol (10 µM) ($n = 2$ biological repeats; 10,000 events; ns $P > 0.05$). **c)** Representative western blot images of viral protein expression in Ov4Carbo cells following treatment with $dl$922-947 (MOI 1) and carvedilol (10 µM) 96-144 hpi. **d, e** Densitometry analysis of viral protein expression, expressed as ratios of **d** E1A/Hsc70 and **e** adenovirus structural proteins (Hexon + Penton + Fibre + core)/GAPDH ($n = 5$ biological repeats; mean ± SEM; paired $t$-test, *$P \leq 0.05$). **f** Viral replication in Ov4Carbo cells following treatment with $dl$922-947 (MOI 1) and carvedilol (10 µM) 96-144 hpi as measured by qPCR for adenoviral Hexon ($n = 6$ biological repeats; mean ± SEM; paired $t$-test, *$P \leq 0.05$). **g, h** Viral titre in Ov4Carbo cells following treatment with $dl$922-947 (MOI 1) and carvedilol (10 µM) 120 hpi (**g**) and 144 hpi (**h**), measured by TCid$_{50}$ (120 hpi: $n = 4$; 144 hpi: $n = 3$ biological repeats, geo mean ± 95% CI; data displayed as technical triplicates from each individual biological repeat assay).

significantly reduced viral IC$_{50}$ in OVCAR4 (50 µM: 2.4-fold, $P = 0.0302$; 100 µM: 7.0-fold, $P = 0.0217$), Ov4Cis (50 µM: 3.5-fold, $P = 0.0085$; 100 µM: 10.2-fold, $P = 0.0093$), and Ov4Carbo cells (50 µM: 4.0-fold, $P = 0.0477$, 100 µM: 10.8-fold, $P = 0.026$) (Fig. 3a). Combination index analysis confirmed that 50 µM and 100 µM alprenolol were both synergistic with viral doses >10 MOI (Fig. 3b). Moreover, alprenolol also significantly reduced viral IC$_{50}$ in colony formation assays in OVCAR4 (50 µM: 6.4-fold, $P = 0.0313$; 100 µM: 19.8-fold, $P = 0.0156$), Ov4Cis (50 µM: 9.6-fold, $P = 0.0313$; 100 µM: 42.3-fold, $P = 0.0156$), and Ov4Carbo cells (50 µM: 6.8-fold, $P = 0.0313$; 100 µM: 43.2-fold, $P = 0.0313$) (Fig. 3c).

Since we had only observed adenoviral synergy with β-arrestin-biased β-blockers and not other drugs that target the β-adrenergic receptor, we pursued β-arrestins as potential mediators of the synergy we had observed. First, we partially inhibited β-arrestins

using pertussis toxin (PTX). PTX is known to prevent the transition of the β1 adrenergic receptor from a Gαs-coupled receptor to a Gαi-coupled receptor, thereby inhibiting β-arrestin-mediated signalling. Since Gα is only found at the β1 and not the β2 adrenergic receptor (Fig. 3d), PTX is expected to only partially inhibit β-arrestin signalling[25]. In keeping with this, PTX pre-treatment partially reversed but did not completely abrogate synergy between carvedilol and $dl$922-947 in Ov4Carbo cells (Fig. 3e and Supplementary Fig. 4b). Next, we simultaneously deleted β-arrestin 1 and 2 using CRISPR-Cas9 gene editing in Ov4Carbo cells. Clones 2 and 5 exhibited complete absence of β-arrestin 1 and 2 protein by western blot and were selected for downstream experiments (Fig. 3f). In these two clones, β-arrestin deletion did not impact overall cell viability and resistance to carboplatin was retained compared to sensitive OVCAR4 cells, although β-arrestin removal did increase sensitivity to both

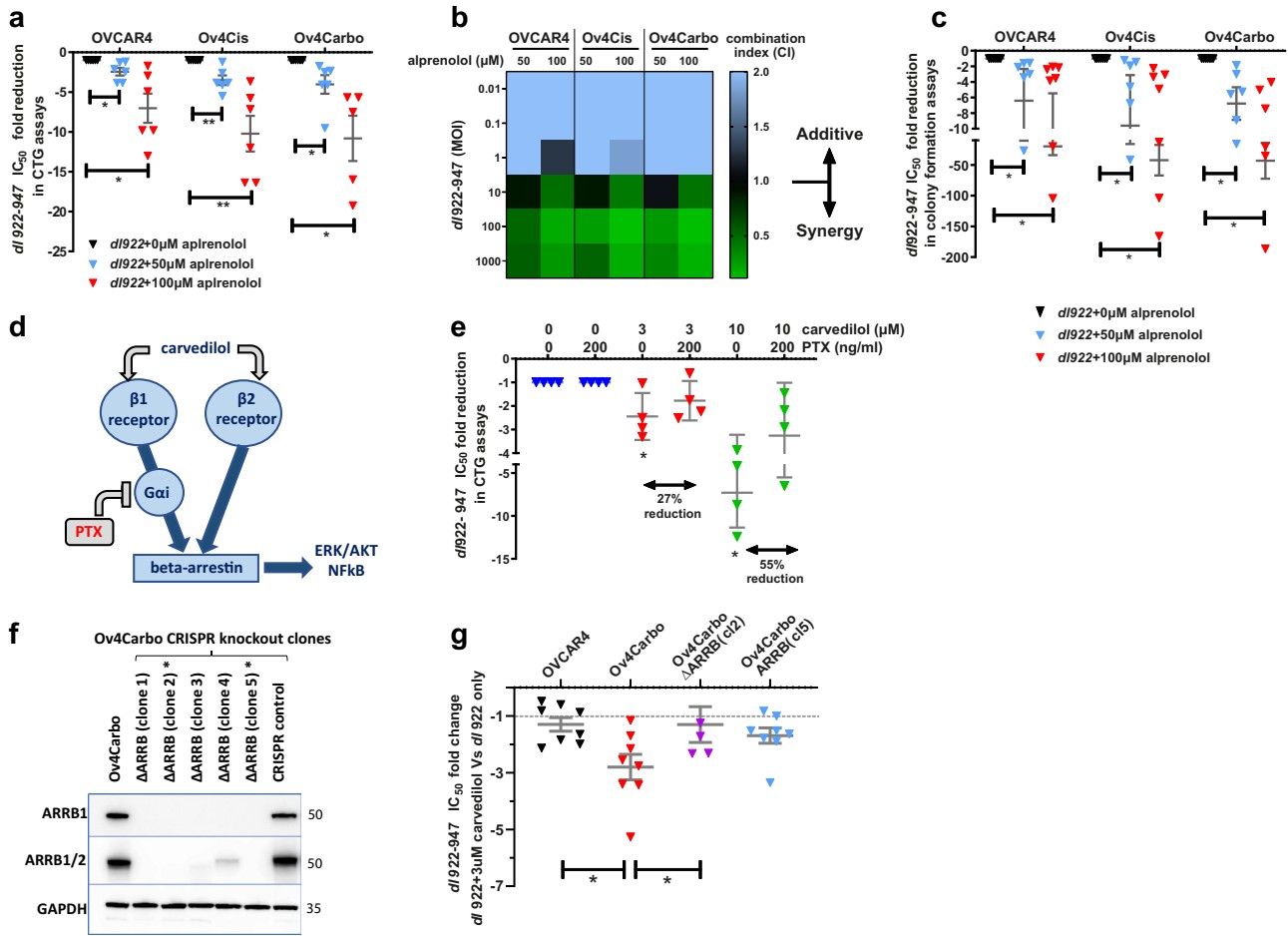

**Fig. 3 Synergy between carvedilol and *dl*922-941 is mediated by β-arrestins. a** Fold reductions of *dl*922-947 IC$_{50}$ in CTG assays following alprenolol treatment compared to virus alone in the OVCAR4 cell line panel ($n = 6$ biological repeats; mean ± SEM; unpaired *t*-test, *$P \leq 0.05$; **$P \leq 0.01$). **b** Heatmap of the combination indexes between alprenolol and *dl*922-947 calculated from ATP-based viability assays. **c** Fold reductions of *dl*922-947 IC$_{50}$ in colony formation assays following alprenolol treatment compare to virus alone in the OVCAR4 cell line panel (+vehicle: $n = 6$; +50 µM: $n = 6$; +100 µM: $n = 7$ biological repeats; mean ± SEM; Wilcoxon matched-pairs signed rank test; *$P \leq 0.05$). **d** Schematic representation of carvedilol-induced β-arrestin signalling. **e** Fold reductions of *dl*922-947 IC$_{50}$ in CTG assays following carvedilol and pertussis toxin (PTX, 200 ng/ml) treatments compared to virus + carvedilol in the OVCAR4 cell line panel ($n = 4$ biological repeats; mean ± SD; unpaired *t*-test; ns $P > 0.05$; *$P \leq 0.05$). **f** Representative western blot of β-arrestin expression in Ov4Carbo arrestin-knockout clones. **g** Fold reductions of *dl*922-947 IC$_{50}$ in CTG assays following 3 µM carvedilol treatment in the Ov4Carbo β-arrestin-knockout clones (OVCAR: $n = 8$; Ov4Carbo: $n = 8$; ΔARRB(cl2): $n = 5$; ΔARRB(cl2): $n = 8$ biological repeats; mean ± SEM; unpaired *t*-test relative to Ov4Carbo, *$P \leq 0.05$).

carboplatin (Supplementary Fig. 6b) and *dl*922-947 (Supplementary Fig. 6c). Our established cell viability combination experiments revealed that carvedilol/virus synergy was significantly reduced in the β-arrestin CRISPR clones compared to unmodified Ov4Carbo cells: CarboΔARRB (cl2); $P = 0.0229$ *vs.* Ov4Carbo and Ov4CarboΔARRB (cl5); $P = 0.0791$ *vs.* Ov4Carbo (Fig. 3g). Together, these experiments strongly implicate β-arrestins as key mediators of the drug/virus synergy we had repeatedly observed.

**Carvedilol and *dl*922-947 influence signalling downstream of β-arrestins.** β-arrestins regulate diverse inflammatory pathways and carvedilol has demonstrated anti-inflammatory activity[36]. Arrestins act as negative regulators of NFκB by protecting the NFκB inhibitor, IκBα, from degradation[37]. Conversely, arrestins promote ERK1/2 activation and thus specific proinflammatory gene expression[38]. Arrestins are also able to modulate NFκB-induced innate immune responses via inhibitory interactions with PI3K, Akt and the mammalian target of rapamycin (mTOR)[26]. Since these

pathways are also relevant in the inflammatory response to viral infection, we considered they could contribute to the synergy we had observed between *dl*922-947 and carvedilol.

To investigate this hypothesis, Ov4Carbo cells were treated with *dl*922-947 (MOI 10), carvedilol (10 µM) or the combination using the same workflow as the initial drug screen (*dl*922-947 or control at 24 h, followed by carvedilol or control at 48 and 120 h). Proteins were harvested over time. Western blot revealed that the combination of *dl*922-947 and carvedilol significantly increased the Akt phosphorylation compared to *dl*922-947 alone (Fig. 4a, b, 96 hpi: $P = 0.0139$ and Supplementary Fig. 4c). Using a luminescence-based NFκB reporter assay, we also found that combined treatment with *dl*922-947 and carvedilol caused a significant, early increase in NFκB activity compared to either agent alone (24 hpi: $P = 0.0208$) (Fig. 4c). Despite the expected anti-inflammatory effect of carvedilol, these data, therefore, demonstrate that the combination of *dl*922-947 and carvedilol paradoxically augmented virus-induced inflammatory signalling in platinum-resistant HGSC cells in vitro.

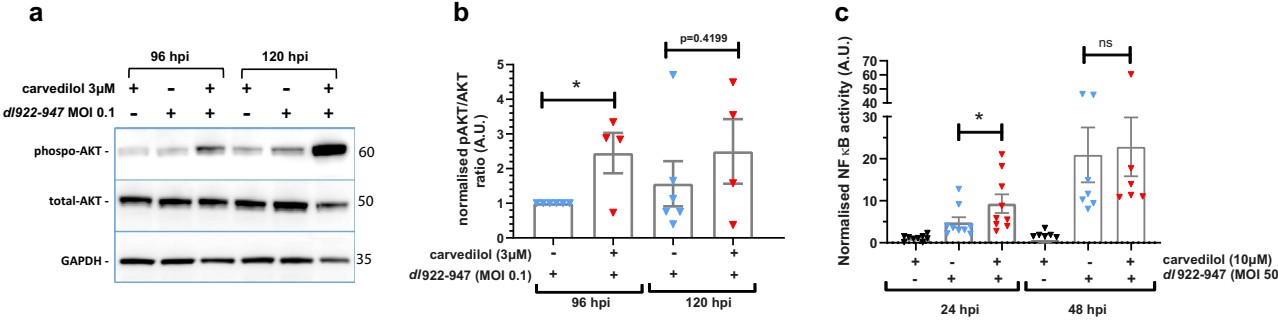

**Fig. 4 Carvedilol and *dl*922-947 influence signalling downstream of β-arrestins. a** Representative western blots of pAkt/Akt in Ov4Carbo cells. **b**) Densitometry ratios of phospho-Akt total Akt in Ov4Carbo cells following infection with *dl*922-947 (MOI 0.1; *n* = 6 biological repeats) and ± carvedilol (3 μM; *n* = 4 biological repeats) 96 and 120 hpi (mean ± SEM, unpaired *t*-test, ns *P* > 0.05, *\*P* ≤ 0.05) **c**) NFκB promoter activity in Ov4Carbo cells following infection with *dl*922-947 (MOI 50) and ± carvedilol (10 μM) 24 hpi (*n* = 9 biological repeats) and 48 hpi (*n* = 7 biological repeats) (mean ± SEM; unpaired *t*-test, *\*P* ≤ 0.05).

**Carvedilol improves the oncolytic efficacy of *dl*922-947 in intraperitoneal murine xenograft models**. To determine the impact of combined carvedilol and *dl*922-947 treatment in vivo, female CD1nu/nu mice were inoculated intraperitoneally (IP) with Firefly luciferase-expressing Ov4Carbo cells. Once tumours were established at day 21 (average radiance between $10^5$–$10^7$ p/s/cm$^2$/sr), mice were randomly allocated to one of four treatment groups: *dl*922-947 ($5 \times 10^7$ viral particles in 200 μl IP), carvedilol (10 mg/kg by oral gavage), the combination or vehicle alone. All treatments were administered daily for five consecutive days. Untreated animals gradually developed invasive tumours together with haemorrhagic ascites resulting in a median survival from tumour cell inoculation to humane end point of 56.5 days (*n* = 28). Carvedilol treatment increased median survival to 78 days (*n* = 18, *P* = 0.0371) but did not prevent the development of tumours and ascites.

*dl*922-947 alone (*n* = 22) and the combination of *dl*922-947 and carvedilol (*n* = 23) both significantly extended survival compared to untreated control mice (Fig. 5a, *P* = 0.0001). Mice in both groups had extremely favourable outcomes with 71.4% and 88.9% of mice respectively surviving for the duration of the experiment thus we did not detect a statistically significant survival difference between these two very effective treatments. Interestingly, only two mice died following combination treatment and both of these deaths occurred very early in the experiment. After these two early deaths, the other 21 mice all survived until the experimental end point 4 months later. One of these mice was culled because of weight loss, rather than the weight gain due to tumour growth and production of ascites that is more typical of this model. At necropsy, this mouse was found to have gastric obstruction due to a small cluster of tumour nodules adjacent to the stomach. The other mouse had a very high baseline luminescence, which was at the upper limit of our pre-defined criteria for enrolment in the study consistent with extensive disease even at this very early time point. Tumour samples were harvested at end point and all available tumours were stained for the proliferation marker Ki67 (Fig. 5b). Analysis of the distribution of Ki67 staining revealed that 20.3 ± 4.7% of the surface area of untreated tumours was positive for Ki67 however this was significantly reduced in carvedilol-treated animals to 10.2 ± 3.1% (Fig. 5c; *P* = 0.0092).

Weekly bioluminescence imaging (BLI) confirmed tumour growth in the untreated and carvedilol-only groups (Fig. 5d–f). In contrast, treatment with virus alone and with the combination of virus and carvedilol effectively controlled tumour growth such that radiance reduced below the baseline level required for initial randomisation in the experiment ($10^5$–$10^7$ p/s/cm$^2$/sr). Importantly, in combination-treated mice, this reduction to baseline

was achieved only 2 weeks following completion of treatment. Tumour control was slower following single-agent *dl*922-947 and the same reduction in radiance took 4 weeks longer to achieve (Fig. 5e). Radiance measurements at week 6 (i.e. 3 weeks after the start of treatment) showed that in *dl*922-947 treated mice, mean tumour-associated radiance was 4.5-fold lower than the untreated group (*P* = 0.016). The reduction in light output was even more marked following combination treatment with carvedilol and *dl*922-947, which significantly reduced mean radiance at week 6 by 13.8-fold (*P* = 0.0043) compared to control-treated animals (Fig. 5f).

**Improved anti-cancer activity following virus and carvedilol co-treatment is associated with induction of innate immunity**. To investigate the mechanisms by which carvedilol augments viral activity, including the in vivo inflammatory response, a further cohort of mice were treated either with *dl*922-947 alone or *dl*922-947 and carvedilol combined. Five mice per treatment group were culled at three pre-defined time points: 1, 3 and 5 weeks after completion of treatment (referred to as time points 1, 2 and 3) (Fig. 6a). Blood and tissue samples were collected from all mice. Tumour tissues were stained by immunohistochemistry (IHC) for PAX8 to identify HGSC tumour, adenoviral protein, macrophages (F4/80) and natural killer cells (integrin-α2; DX5) (Fig. 6b: representative images taken from a mouse treated with the combination of *dl*922-947 and carvedilol and culled at time point 3). Adenoviral protein expression was found to increase over time, and in line with our in vitro data (Fig. 2c), tumour samples from the combination treatment group showed greater adenovirus expression compared to those treated with virus alone. This was most marked at the earliest time point but persisted throughout the experiment (median value increased 6.94-fold, 3.86-fold and 2.08-fold at time points 1, 2 and 3 respectively; Fig. 6c).

Since our in vitro data indicated that carvedilol could modify the inflammatory response to oncolytic *dl*922-947 (Fig. 4), we investigated inflammatory cells in murine tumours. Tumour infiltration with monocytes/macrophages (F4/80) and NK cells (DX5) generally increased over time (Fig. 6c) and by time point 3, both were higher in all virus-treated mice compared to tumour samples previously harvested at end point from mice that had received carvedilol or vehicle alone (depicted in Fig. 5). Combined treatment with *dl*922-947 and carvedilol increased tumour infiltration by macrophages and NK cells compared to virus treatment alone. In keeping with the temporal changes we observed in intratumoural adenoviral proteins, this inflammatory infiltrate was also most noticeable 1 week after completion of

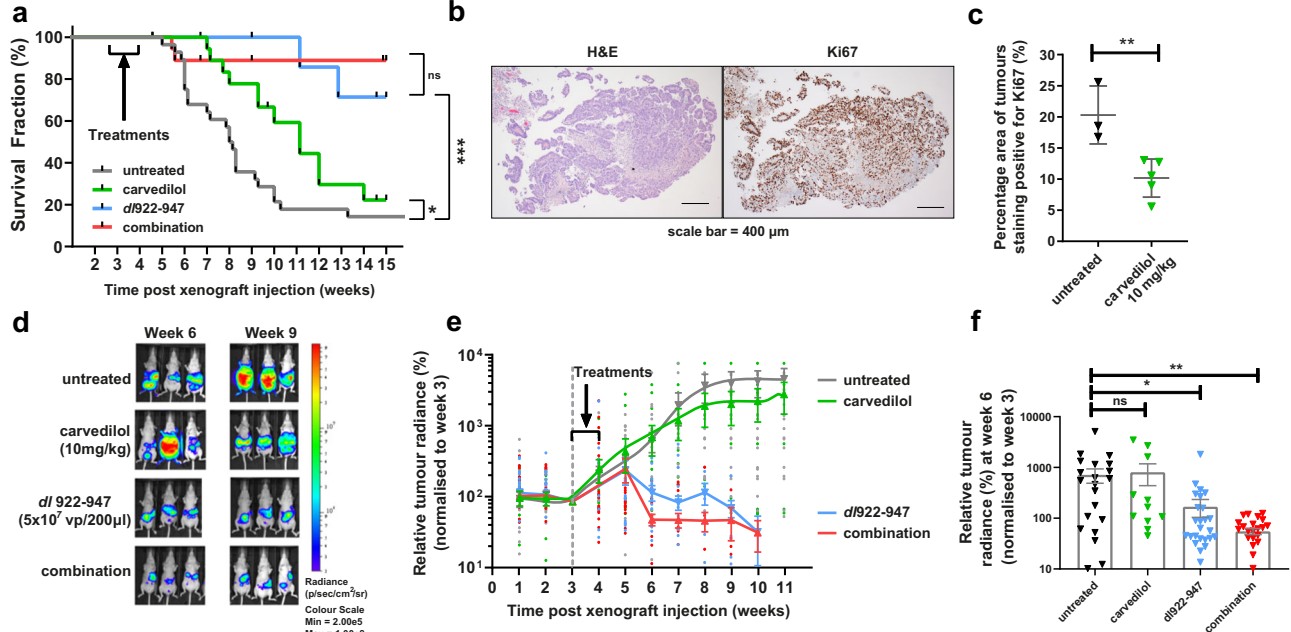

**Fig. 5 Carvedilol improves the oncolytic efficacy of *dl*922-947 in intraperitoneal murine xenograft models. a** Kaplan and Meier survival curves of Ov4Carbo tumour-bearing mice treated with vehicle (n = 28), carvedilol (10 mg/kg, n = 18), *dl*922-947 (5 × 10⁷ vp/200 μl, n = 22) and the combination (n = 23) (from three biological repeat experiments; Log-rank (mantel-Cox) test; ns P > 0.05; *P ≤ 0.05; ***P ≤ 0.0001). **b** Representative histology images of tumour samples stained with H&E and Ki67 (scale bar: 400 μm). **c** Percentage of total surface area of tumour samples treated with vehicle (n = 3) and carvedilol (10 mg/kg, n = 5) staining positive for Ki67 (mean ± SD, unpaired t-test, **P ≤ 0.01) **d** Representative images of luciferin radiance at weeks 6 and 9. **e** Tumour-associated luciferin radiance over time normalised to baseline readings (week 3) prior to treatments with vehicle (n = 28), carvedilol (10 mg/kg, n = 18), *dl*922-947 (5 × 10⁷ vp/200 μl, n = 22) and the combination (n = 23) (from three biological repeat experiments, mean ± SEM). **f** Tumour-associated luciferin radiance at week 6 (vehicle n = 24, carvedilol n = 11, *dl*922-947 n = 28, combination n = 25 mice from three repeat experiments, mean ± SEM, unpaired t-test, ns P > 0.05, *P ≤ 0.05**, P ≤ 0.01).

treatment (fold increases of 2.17-fold and 2.64-fold respectively) (Fig. 6c).

To further explore the influence of carvedilol on the systemic inflammatory response, a panel of 19 circulating murine cytokines was quantified in murine blood at necropsy using multiplexed electrochemiluminescent analysis (Meso Scale Diagnostics®). From the panel of 19 cytokines, six were undetectable in our samples (IL-2, IL-4, IL-9, IL-12p70, IL-15 and IL-17a) and 7 seven did not change over time or according to treatment group (IL-1b, IL-5, IL-6, IL-10, IL-33, KC/GRO, MIP2) (Supplementary Data 4). The remaining six cytokines are shown in Fig. 6d. At time point 1, TNFα was significantly elevated (4.1-fold increase, P = 0.0392) in the combination-treated mice compared to mice treated with *dl*922-947 alone (Fig. 6d). Five other circulating cytokines were also more elevated in the combination-treated group: IFNγ (6.9-fold increase, P = 0.233), IL-27/28 (3.6-fold increase, P = 0.1886), MCP-1 (20.6-fold increase, P = 0.1553), IP-10 (3.3-fold increase, P = 0.1291) and MIP1α (5.8-fold increase, P = 0.2544) (Fig. 6d). These differences were observed 1 week after completion of treatment (time point 1) and for certain cytokines (IFNγ, TNFα and IL-27/28) persisted until time point 3 (Fig. 6e).

In summary, the combination of carvedilol and oncolytic adenovirus enabled rapid clearance of platinum-resistant intraperitoneal HGSC associated with early induction of a local inflammatory cell infiltrate and systemic release of multiple inflammatory and antiviral cytokines.

## Discussion
Our study demonstrates, that β-arrestin-targeted therapies such as carvedilol can improve the efficacy of oncolytic viruses in

HGSC. Carvedilol is already widely used for the treatment of cardiac disease and so could be introduced into clinical trials as a safe and cost-effective adjunct to viral therapies. Interestingly, improved ovarian cancer-specific survival has previously been demonstrated in epidemiological studies of patients receiving non-specific β-blockers for other indications. In one study, 344 HGSC patients receiving a range of β-blockers had a better overall survival of 90 compared to 38.2 months[32]. In another study of ovarian cancer patients over the age of 60, the use of non-selective β-blockers (151 patients) was again associated with better survival (hazard ratio = 0.579)[31]. It has been postulated that this could be due to the known anti-proliferative effects of these drugs[39–41] but both studies group multiple different β-blocking drugs in their analysis and provide no detail on the specific effect of carvedilol. Moreover, we are not aware of any studies describing the anti-cancer activity of carvedilol in human patients. Interestingly, we did observe a survival advantage following single-agent carvedilol treatment in our murine xenografts. The known anti-proliferative effects of carvedilol could potentially explain this observation[39,41] and indeed Ki67 expression within tumour tissues was reduced in carvedilol-treated animals.

Our most impactful finding, however, was that the combination of carvedilol and oncolytic adenoviruses had impressive anti-cancer activity even in platinum-resistant intraperitoneal models. Combination therapy was able to rapidly eliminate tumour-associated bioluminescence, only 2 weeks following completion of treatment. Single-agent *dl*922-947 did eventually achieve tumour control but this reduction to baseline radiance took 4 weeks longer than the combination of *dl*922-947 and carvedilol. Thus although combination treatment controlled tumours more rapidly, both the combination and *dl*922-947 alone were ultimately highly effective. This was reflected in the impressive

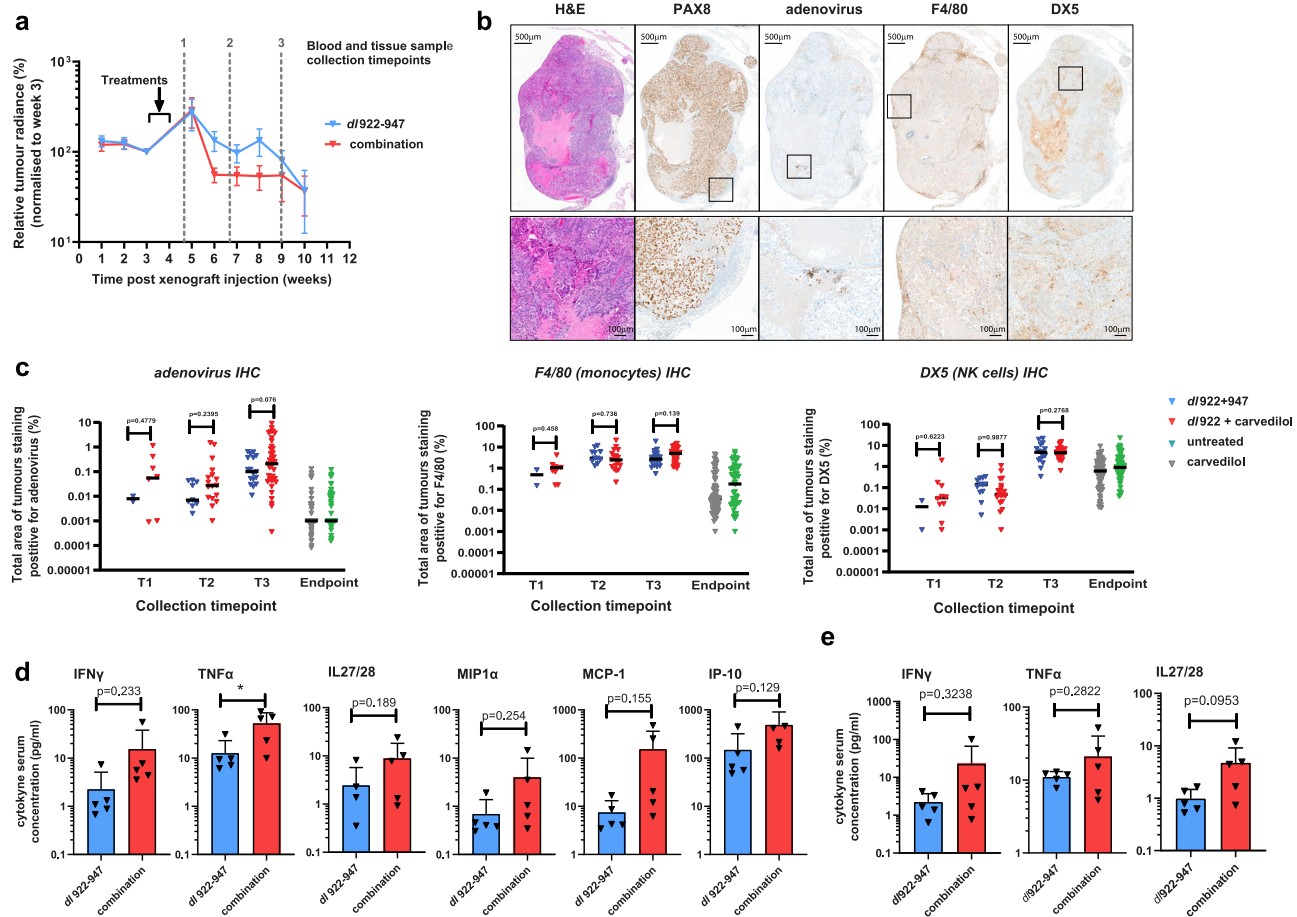

**Fig. 6 Co-treatment with virus and carvedilol induces innate immunity in vivo. a** Inset of radiance data (shown with individual data points in Fig. 5e) to indicate experimental plan and time points for collection of blood and tissue samples in remainder of this figure. **b** Representative histology images of (top) tumour samples stained for H&E, PAX8, adenovirus, F4/80 and DX5 (scale bar: 500 μm), (low) magnified area denoted by the box above (all images are from a single mouse treated with the combination of dl922-947 and carvedilol and culled at time point 3; scale bar: 100 μm). **c** Percentage of the total surface area of tumour samples staining positive for adenovirus (left panel), F4/80 (macrophages, middle panel) and DX5 (NK cells, right panel) (frequency distribution violin plot, median (black) + interquartile range (grey); n = 5 mice per group, mean 28 measurements per group). **d** Systemic cytokines quantified at time point 1 in mice treated with carvedilol and dl922-947 (n = 5 mice, mean + SD, unpaired t-test, ns P > 0.05, *P ≤ 0.05). **e** Systemic cytokines quantified at time point 3 in mice treated with carvedilol and dl922-947 (n = 5 mice, mean + SD, unpaired t-test, ns P > 0.05).

survival benefit seen with both treatments. A much greater number of mice would be required to detect a statistical difference between the two groups.

Our mechanistic investigation of combined treatment with carvedilol and oncolytic adenovirus, demonstrated that β-arrestin signalling underpinned the synergy we observed. G protein-coupled receptors and β-arrestin driven pathways are not specific to HGSC, raising the exciting possibility that this combination therapy could also prove effective in other cancer types. Equally our data confirming that carvedilol synergises with wild-type adenovirus imply that similar synergy may also occur with a range of other oncolytic adenoviruses. Carvedilol and alprenolol are both classified as β-arrestin-biased-non-selective β-blockers and as such are a distinct subclass of β-blocking agents[35]. Both drugs exhibit additional activities including alpha antagonism and although none of the nine alpha blockers included in our drug screen showed synergy with adenoviruses, we cannot discount the possibility that additional signalling pathways could contribute to the synergy we observed.

β-arrestins are known to regulate antiviral inflammatory responses. Our data show that in vitro treatment with dl922-947, followed by targeting of β-arrestins by carvedilol, was able to induce downstream activity of Akt and NFκB. In vivo, simultaneous

treatment with dl922-947 and carvedilol induced a local inflammatory cell infiltration together with systemic release of inflammatory and antiviral cytokines including IFNγ, TNFα and IL-27/28. This inflammatory response could represent either a cause or a consequence of increased oncolytic activity and interestingly our findings appear to contrast with current literature suggesting that carvedilol is associated with reduced inflammatory markers in disease models such as diabetes and acute pancreatitis[36]. A possible explanation is that the known anti-inflammatory effect of carvedilol promoted viral replication and experiments comparing different scheduling of the two agents could be used to further explore this possibility.

Oncolytic viruses hold promise as a novel means to overcome immune evasion in cancers and have been modified to elicit a favourable antitumour immune response in otherwise immune-silent cancer cells[42–44]. This is predominantly achieved through the release and presentation of antigens following virus-induced cell death. Adenovirus-induced cell death, however, is a multi-factorial process including autophagy, apoptosis, necrosis and pyroptosis in addition to induction of an immune response[45–49]. Further studies to determine the mechanism of cell death following the combination of carvedilol and dl922-947 would likely help to unravel the relationship between the improved

viral oncolysis and the altered inflammatory response that we observed.

T cells are known to play a key role in the antitumour efficacy of oncolytic viral agents[50] and this has been elegantly described following T-VEC[51]. An important limitation of our study is that we were unable to interrogate this aspect of oncolytic activity in the immunodeficient xenograft models used here. Adenoviruses are species-specific, so human adenoviruses replicate poorly in non-human cells although it has been possible to achieve modest intratumoural replication in specific murine cell lines[52], following viral modification[53] and with certain combination therapies[54]. Since dl922-947 is not armed, its antitumour effect is entirely reliant on its ability to specifically replicate within human tumours[19]. The resulting amplification of viral dose is advantageous in the treatment of disseminated malignancies like the clinically representative intraperitoneal models of HGSC we used here. T-cell immunity induced by oncolytic adenoviruses has been modelled in immunocompetent tumour-bearing mice but only following direct intratumoural injection of non-replicating adenoviruses into localised tumours[50,55,56]. Adenoviral species-specificity means that it has not yet been possible to evaluate the role of T cells in the antitumour response to replicating oncolytic adenoviruses, particularly following systemic delivery to mice with disseminated cancer. The recent development of humanized mice in which immunodeficient mice are engrafted with peripheral blood mononuclear cells or haematopoietic stem cells (reviewed in ref. [57,58]), may in future offer a solution to this pervasive challenge. Potentially, an ovarian cancer intraperitoneal xenograft could be created in such a mouse but we are not aware of any published or commercial evidence that such an ovarian cancer model has been generated to date. Investigating the combination of carvedilol and oncolytic dl922-947 in an immunocompetent setting is an obvious next step and since carvedilol is a comparatively safe drug, this could realistically include clinical trials.

In conclusion, here we report that carvedilol can be repurposed to enhance the anti-cancer efficacy of oncolytic viral therapies. Our study also confirms the β-arrestin pathway as a druggable target to enhance viral efficacy. The combination of carvedilol with oncolytic adenoviruses is a cost-effective and safe approach that could now be rapidly translated for patient benefit.

## Materials & methods

**Cell lines, drugs and viral constructs.** Human cell lines OVCAR4 and Cov318 were used because they have previously been classified as highly representative of human HGSC[59]. Both were obtained from Prof F Balkwill (Barts Cancer Institute, UK). Platinum-resistant HGSC cell lines were generated by serial culture in increasing concentrations of either cisplatin (Ov4Cis, CovCis) or carboplatin (Ov4Carbo) and then subsequently cultured without drug and used at low passage (<10) as we previously described[60]. Cells were cultured at 37 °C and 5% $CO_2$ in DMEM (Lonza) supplemented with 10% FBS (Gibco) and 1% penicillin/streptomycin (10,000 units/ml; Gibco). Human embryonic kidney cells JH293 and HEK293 were used for viral production and for $TCID_{50}$ assays. All cell lines underwent 16 locus STR verification (DNA Diagnostics Centre, London, UK: June 2015-February 2016 and European Collection of Authenticated Cell Lines August 2019) and weekly mycoplasma testing.

The following drugs and compounds were used: Carvedilol (Cayman Chemicals), Alprenolol HCL (Sigma-Aldrich), Propranolol HCL (Cambridge Bioscience), Isoprenaline (Sigma-Aldrich), Pertussis toxin from Bordetella pertussis (Sigma-Aldrich). Human adenovirus five wild-type (AD5), dl922-947 and AD5-eGFP were all originally obtained from Prof. Y. Wang (Barts Cancer Institute,

UK). dl922-947 is an AD5 mutant with deletions in E1A-CR2 and E3B[19]. AD5-eGFP is a wild-type AD5 expressing green fluorescent protein (GFP).

**Drug screen.** To screen HGSC cell lines for existing drugs that synergise with dl922-947, we used a 96-well cell viability assay based on our previous work[61,62]. Our library includes 1177 small molecules (Selleck Chemicals), in which 90% are marketed drugs, and the remaining 10% are bioactive alkaloids. OVCAR4 cells (platinum-sensitive) were compared to either Ov4Cis (cisplatin-resistant) or Ov4Carbo (carboplatin-resistant) in two separate experiments. Cells were plated in 96-well plates ($10^3$ cells per well). After 24 h, cells were treated with either dl922-947 (MOI (multiplicity of infection) of ten plaque-forming units (pfu) per cell) or vehicle control. Then at 48 h and again at 120 h, cells were treated with the library (10 μM based on our previous work[61,62]) or dimethyl sulphoxide (DMSO) vehicle control. Each drug in the compound library was therefore tested alone and in combination with dl922-947 in all three cell lines, in one well of the 96-well plate per experimental condition.

Cell viability was assessed at 192 h (i.e. 72 h after the second library drug treatment) using a luciferase-based ATP assay (CellTitre-Glo, Promega) according to the manufacturer's instructions. Luminescence output was quantified with a PerkinElmer plate reader. For each well in the 96-well plate, the effect of treatment was represented as a log2-surviving fraction (s.f.) and expressed relative to the median s.f. for the entire 96-well plate. In each cell line, s.f. was compared between library-treated and vehicle-treated wells and also between library + dl922-947 and dl922-947-treated wells. Results from the two drug screens are shown in Supplementary Data 1 (OVCAR4 and Ov4Cis) and Supplementary Data 2 (OVCAR4 and Ov4Carbo). To identify selective ability to synergise with dl922-947, we used our established cut-off of s.f. < −2 following library + dl922-947 together with s.f. > −2 for library compound alone[62]. Using these criteria, six compounds were identified as potential hit drugs in both Ov4Cis and Ov4Carbo cells (Supplementary Data 3).

**Cell viability, synergy validation, and colony forming assays.** Cell viability and synergy validation experiments were conducted by plating cells in 96-well plates ($10^3$ cells/well) in triplicate wells per condition. In synergy validation experiments cells were treated using the same experimental protocol as the initial compound library screen (see above) but using drug doses ranging from 1–10 μM and viral doses between MOI 0.01–1000 pfu/cell. Cell viability was again measured using CellTitre-Glo 72 h after the second drug treatment. In β-arrestin inhibition experiments, cells were pre-treated for 24 h with 200 ng/ml pertussis toxin prior to carvedilol treatment.

Colony formation assays were conducted by plating $10^3$ cells in duplicate in 6-well plates. After 24 h incubation, cells were treated simultaneously with virus (MOI 0.01–100 pfu/cell) and drug (1–10 μM). Repeat treatments were administered twice weekly until distinct colonies formed (day 10–14). Colonies were fixed in methanol and stained with sulphorhodamine B. Colonies were imaged using an Amersham Imager 600 (GE Healthcare) and counted using the ImageQuant TL software package.

In all cases, at least three biological repeat experiments were conducted. Dose-response curves were generated with GraphPad prism v7.04 (nonlinear regression fit to a five-parameter equation) to determine $IC_{50}$. Drug/virus synergy was determined according to the Chou–Talalay method and calculated using the CompuSyn V.1 software package (non-constant ratio) to generate isobolograms where a CI (combination index) > 1 indicates synergy, CI = 1 indicates addition, and a CI < 1 indicates antagonism[34].

**Viral assays**. Virus infectivity was assessed using a wild-type adenovirus expressing eGFP under the control of the CMV promoter. Cells were infected with AD5-eGFP (MOI 10) and simultaneously treated with carvedilol (10 μM). GFP fluorescence was quantified 24 h later on an LSRFortessa (BD Biosciences). All conditions were repeated in triplicate and analysed using FlowJo software.

Virus binding was determined by infecting $2 \times 10^5$ cells (in triplicate) with $5 \times 10^3$ viral particles (vp) in 1% BSA/PBS at 4 °C for 1 h with moderate shaking. Cells were then washed 3x with cold 1% BSA/PBS. DNA was isolated using a QIamp DNA Blood Mini Kit (Qiagen) according to the manufacturer's instructions. Real-time PCR was performed on an ABI Prism 7500 (Applied Biosystems) using Taqman Gene Expression Assay with the following conditions: 25 °C, 10 min; 37 °C, 120 min; 85 °C, 5 min). DNA for the adenoviral structural protein, hexon (5′: GGTGGCCATTACC TTTGACTCT; 3′: GGGTAAGCAGGCGGTCATT; 6-Fam probe: CTGTCAGCTGGCCTGG) was quantified by qPCR against a standard curve of viral particles to quantify adenovirus genome copy number per ng of DNA.

For the viral replication assay (TCID50), cells were infected with *dl*922-947 (MOI 10) and treated with carvedilol (10 μM) as before (see synergy validation above). Cells were harvested at 144, 168 and 192 h in 0.1 M Tris pH8.0 and underwent three rounds of freeze/thawing. All conditions were collected in triplicate and analysed by limiting dilution method (TCID$_{50}$) on JH293 cells in triplicate experiments and end point titres were calculated[63].

**Western blotting**. Protein lysates were collected in ice-cold lysis buffer (150 mM NaCl, 50 mM Tris, 0.05% SDS, 1% triton) and quantified using a Pierce BCA assay kit (Thermo Scientific). Protein samples (20–30 μg) underwent routine electrophoresis on precast 4–12% Bis TRIS pre-cast gels (Invitrogen) and were transferred to nitrocellulose membranes (GE Healthcare). Blots were developed using enhanced chemiluminescence (GE Healthcare) and imaged with an Amersham Image 600 (GE Healthcare). Densitometry analysis was performed using Image-Quant TL software. E1A: Santa Cruz sc-430, 1:1000; Adenovirus: Abcam ab36851, 1:5000; Hsc70: Abcam ab36851, 1:10000; GAPDH: Santa Cruz sc-47724, 1:10000; phospho-Akt: Cell Signalling 9271L, 1:1000; Akt: Cell Signalling 9272; phospho-ERK1/2: Cell Signalling 4370, 1:1000; ERK1/2: Cell Signalling 9102; ARRB1/2: Cell Signalling 4674S, 1:500; ARRB1: Cell Signalling 12697S, 1:500.

**NFκB reporter assay**. NFκB activity was measured using a Dual-Luciferase® Reporter Assay System (Promega). Firefly luciferase was under the control of the NFκB consensus binding site and Renilla luciferase under an SV40 ubiquitous promoter. The plasmids were amplified using NEB alpha5 competent cells in LB broth (100 μg/ml Ampicillin) and purified using a Monarch® Plasmid Miniprep Kit (NEB). Cells were transfected using FuGENE® HD Transfection Reagent (Promega). Following transfection, cells were treated using the same workflow as synergy validation experiments (see above). At all time points, samples were collected in 1x passive lysis buffer according to the manufacturer's instructions and stored at −80 °C prior to quantification of light output using a PerkinElmer plate reader.

**CRISPR-Cas9 gene editing**. Simultaneous CRISPR-Cas9 deletion of both β-arrestin 1 and β-arrestin 2 was performed in Ov4Carbo cells using the Dharmacon™ Edit-R™ Cas9 nuclease protein and synthetic guide RNA transfection protocol. crRNA details and target sequences were as follows: ARRB1: CM-011971-01-00 02, ATCTCAAAGAGCGGAGAGGT; CM-011971-02-0002, GT

CTGCATACTGGCGCACTA; CM-011971-03-0002, GCTTCT CTCGGTTATTGGCT; ARRB2: CM-007292-01-0002, CCAG GTCTTCACGGCCATAG; CM-007292-02-0002, GACTACCTG AAGGACCGCAA; CM-007292-03-0002, GAGAAACCCGGG ACCAGGTA. Cas9 transfection was achieved using Edit-R hCMB-PuroR-Cas9 expression plasmid (U-005100-120), Edit-R CRISPR-Cas9 Synthetic tracrRNA (U-002005-20) and Dharma-FECT Duo Transfection Reagent (T-2010-02). Cells were incubated with transfection reagents for 6 h and puromycin selection was performed 24 h later. Single-cell colonies were selected by serial dilution. CRISPR deletion of β-arrestin 1 and β-arrestin 2 was confirmed by western blot

**In vivo assays**. Experiments were conducted following approval from the internal Institutional Review Board at Barts Cancer Institute, Queen Mary University of London. The work was conducted under project licence P1EE3ECB4, as required by the UK government Home Office in accordance with the Animals (Scientific Procedures) Act 1986 (ASPA). Platinum-resistant Ov4Carbo cells were modified using lentiviruses to express dual RFP/Luciferase reporter as we previously described[60]. Cancer cells were inoculated by intraperitoneal (IP) injection into 6-week old female CD1*nu/nu* mice (Charles River Laboratories) in sterile PBS ($5 \times 10^6$ cells/200 μl). Tumour growth was monitored by weekly bioluminescence imaging. Animals received IP injections of D-Luciferin monopotassium salt (ThermoFisher, 3.75 mg/200 μl PBS) and light emission was recorded on an IVIS® Spectrum In Vivo Imaging System (PerkinElmer). Tumours were allowed to become established and only mice exhibiting an average radiance between $10^5$–$10^7$ p/s/cm$^2$/sr by day 21 post inoculation were included in the studies (34 mice excluded on this basis from a total of 152). Mice were randomly allocated to treatment groups and treatments were administered by IP injection of *dl*922-947 ($5 \times 10^7$ vp in 400 μl PBS) and by oral gavage of carvedilol (10 mg/kg in PBS + 1% methylcellulose). Treatments were administered once daily for 5 days on day 21–26 and in all cases, vehicle controls were administered by the same route. The administration schedule of daily treatment for five days is based on our previously published work in different murine models[11,21] although this time we used a lower dose of $5 \times 10^7$ vp to detect an incremental benefit with the addition of carvedilol. The carvedilol dose is based on publications using between 5 and 15 mg/kg daily[56,64,65] and since it is an oral drug, we administered carvedilol to mice by oral gavage. Our animal license restricted us to five carvedilol treatments and so we elected to administer carvedilol on the same days as *dl*922-947 to reflect the usual management of human patients, in which combination therapies are most commonly given on the same treatment day. Researchers were blinded to the treatment groups. Tumour growth and animal weight were monitored until they reached humane end points (defined by UK Home Office Regulations). Minimum sample sizes were calculated using the 'sample size–survival analysis' software (www.sample-size.net).

**Histology**. Tissue samples were fixed in 10% Formalin and embedded in paraffin wax (Leica RM2255 microtome). Sections were cut in Palm slides and underwent Haematoxylin&Eosin (H&E) staining and automated Immunohistochemistry staining (Leica ST5010). Histology slides were imaged on a Pannoramic 250 Slide Scanner (3DHISTECH) and images exported to ImageJ. Images were converted to RGB stacks and a threshold was set across all images to identify positive staining. H&E and PAX8 staining was used to identify HGSC tissue. Regions of interest (ROIs) were then manually drawn around discrete areas of HGSC tissue. The data were reported as the total area percentage

of the ROI staining positive. PAX8: Abcam ab13611, 1:100; F4/80: Biorad MCA497GA, 1:1000; Adenovirus: Abcam ab8251, 1:1000; DX5: Abcam ab133557, 1:250.

**Cytokine analysis**. Murine cytokine concentrations were determined in serum samples isolated by cardiac puncture at the experimental end point. Cytokines were detected in duplicate wells per sample using the V-PLEX Proinflammatory Panel 1 Mouse Kit and V-PLEX Cytokine Panel 1 Mouse Kit (Meso Scale Diagnostics, LLC®) according to the manufacturer's instructions. Data were analysed using DISCOVERY WORKBENCH® 4.0 software.

**Statistics and reproducibility**. All data are presented as mean ± s.d. and statistical analysis was performed using GraphPad Prism v8.0. Normal distribution was tested using the Shapiro–Wilk normality test. Statistical significance was calculated using a two-tailed unpaired $t$-test unless otherwise specified ($*P < 0.05$; $**P < 0.001$; $***P < 0.0001$). $n$ refers to biological replicates and the established scientific standard of $n \geq 3$ was applied throughout. Dose-response curves and $IC_{50}$ values were calculated using GraphPad Prism v.8.0 (nonlinear regression fit to a five-parameter equation). Synergy was determined according to the Chou–Talalay method (non-constant ratio) and calculated using the CompuSyn V.1 software package.

**Reporting summary**. Further information on research design is available in the Nature Research Reporting Summary linked to this article.

## Data availability

Source data associated with this project are available in Supplementary Data 1–4 and deposited in figshare: (https://figshare.com/articles/figure/Carvedilol_targets_arrestins_to_rewire_innate_immunity_and_improve_oncolytic_adenoviral_therapy/17013317)[66].

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

## Acknowledgements

M.L., J.I.H., S.M. and F.N. were supported by ML's Cancer Research UK Advanced Clinician Scientist Fellowship (C41405/A19694). M.L. and J.I.H. received support from the Barts Cancer Institute Impetus Fund. EAoS was funded by an MRC-DTP PhD studentship. M.L. and A.B. received support from ML's Cancer Research UK Clinician Scientist Fellowship (C41405/A13034). M.L. and J.S. were supported by a Barts and The London Charity Strategic Research Grant (467/2244). We would like to acknowledge Prof. Fran Balkwill for kindly providing us with the human HGSC cell lines OVCAR4 and COV318 and Dr Ashley Browne for creating the derived resistant cells. To Dr Kunal Shah for help with NFκB assay and CRISPR experiments and to Dr Rathi Gangeswaran for help setting up the viral attachment assays. This work was supported by a Cancer Research UK Centre Grants: C16420/A18066 and C355/A25137. We specifically thank core services at the Barts Cancer Institute including the Animal Technician Service, the Biological Services Unit, microscopy, imaging and histology.

## Author contributions

J.I.H.: conceptualisation, methodology, data curation, formal analysis, writing original draft, writing–review and editing. B.O., E.A.o.S., A.B., N.C., S.M., F.N., J.S.: methodology, data curation, formal analysis. S.A.M.: conceptualisation, methodology, formal analysis, project supervision, writing—review and editing. M.L.: conceptualisation, methodology, data curation, formal analysis, writing original draft, writing—review, editing, supervision, funding acquisition and project administration.

## Competing interests

The authors declare no competing interests.
