## [Peer Review File · Communications Biology]

Reviewers' comments:

Reviewer #1 (Remarks to the Author):

In the present manuscript, Hoare and colleagues reported that carvedilol, a β -arrestin-biased β -blocker, exhibited synergistic anti-ovarian cancer effects with oncolytic adenovirus. The study revealed that carvedilol augmented the activity of oncolytic adenoviruses via β -arrestins to re-wire cytokine networks and innate immunity. This study is interesting. I have a few major and some minor comments:

Major issues:

1. The authors reported that carvedilol increased virus-induced cell death by promoting viral life cycle. Carvedilol and dl922-947 showed a good synergistic effect in vitro. However, combination therapy with carvedilol and dl922-947 did not show superior survival benefit compared with dl922-947 monotherapy (Figure 5A). Moreover, at weeks 10, the relative tumor radiance of the combination therapy group was not significantly different from that of the dl922-947 monotherapy group (Figure 5E). Please comment.
2. The study showed carvedilol co-treatment was associated with induction of innate immunity. Relevant study data were obtained from immunodeficient xenograft models. Nude mice lack T cells. A large number of studies have confirmed that T cells play an important role in oncolytic virus mediated antitumor effects. Therefore, the synergistic effects and mechanisms of combination therapy should be further verified in an immunocompetent murine model.
3. Mice were administered by IP injection of dl922-947 (5×10^7 vp in 400 μ l PBS) and by oral gavage of carvedilol (10mg/kg in PBS +1% methylcellulose). Treatments were administered once daily for 5 days. What is the basis for the therapeutic schedule?

Minor issues:

1. In the title of the legend of Figure 1, "dl922-941" should be changed to "dl922-947".
2. In Figure 6F, the relative tumor radiance of combination therapy group and dl922-947 monotherapy group should be statistically analyzed.

Reviewer #2 (Remarks to the Author):

The authors previously identified the oncolytic virus dl922-947 as a potential antitumoral agent against ovarian cancer. Here they performed a drug screening to identify agents able that could synergize with virotherapy and counteract the growth of chemoresistant ovarian cancer cells. Their screening identified the β -adrenergic receptor (β -AR) antagonist, carvedilol, as a potential agent that could act in synergy with dl922-947. They performed a series of straightforward, well-thought and well-conducted experiments and indeed confirmed that carvedilol synergized with dl922-947 in ovarian cancer cell lines (especially in those that were chemoresistant) and promoted viral replication. Likely, carvedilol effect was due to the beta-arrestin pathway as proved by CRISPR-mediated KO of beta-arrestin and pertussis toxin treatment, which both reduced carvedilol/virus synergy.

Notably, carvedilol also improved dl922-947 oncolytic efficacy in intraperitoneal murine xenografts with induction of inflammatory infiltrate and release of inflammatory cytokines consistent with the induction of an antitumoral innate immune response.

Overall, the authors identified a new potential combination strategy that could prove useful for the treatment of aggressive chemoresistant ovarian cancer (and likely translatable to other tumors). I support the publication of this study and I do not have major issues to flag. I just have a couple of minor comments, listed below:

-On page 19 lines 397-399, the author state: "Despite the expected anti-inflammatory effect of carvedilol, these data therefore demonstrate that the combination of dl922-947 and carvedilol paradoxically augmented virus-induced inflammatory signalling in platinum-resistant HGSC cells in vitro." Could the anti-inflammatory effect of carvedilol favour viral replication? I wonder whether the sequential administration (rather than concomitant use of carvedilol/virus) could help address this issue. The authors might consider discussing this possibility.

- Supplementary tables 1 & 2: please clarify why some plates/ plate positions are marked in red

and provide a color legend

Reviewer #3 (Remarks to the Author):

The manuscript entitled "Carvedilol targets β -arrestins to rewire innate immunity and improve oncolytic adenoviral therapy" by Hoare et al. describe an interesting drug-repurposing approach that combines carvedilol with adenoviruses. This study shows a synergic combination of both the agents resulting in increased oncolytic activity (increased DNA replication, vira protein expression, etc.) mediated by β -arrestins. The efficacy of the combined treatment was demonstrated also in a mouse model, the enhanced anti-cancer activity was accompanied by innate immune cell infiltration and cytokine release.

The manuscript is well organized, the technical procedures and results are accurately described, however I would suggest improving the discussion, in which I find some issues that might be better addressed.

The authors declare that observed a survival advantage following single agent carvedilol treatment (lines 481-482), has the anticancer activity of carvedilol alone been already described? If so, please discuss and add appropriate references.

The authors describe the mechanism of action of the combination treatment that is mediated by β -arrestins, however I would discuss how the anticancer activity of adenovirus might be elicited, i.e known or recently reported cell death mechanisms reported in literature that might account for cancer cell survival decrease or recall of immune cells (apoptosis, immunogenic cell death, etc.). Finally I think that the discussion should be improved.

Editor's Comments:

'...if possible, it would be beneficial if you could look at the effects of the proposed combination therapy in a immunocompetent mouse model (as suggested by Reviewer 1)'

Although we agree that such an experiment could yield relevant results, it is our strong opinion that substantial logistical and financial challenges unfortunately preclude us from performing such a study in the context of the current manuscript. We discuss this at length in our reply to Reviewer 1's comments below.

Reviewers' Comments:

Reviewer #1 (Remarks to the Author):

We are pleased that this reviewer found our study interesting. We have addressed their major and minor comments below.

Major:

1. *'...Carvedilol and dI922-947 showed a good synergistic effect in vitro. However, combination therapy with carvedilol and dI922-947 did not show superior survival benefit compared with dI922-947 monotherapy (Figure 5A).'*

This reviewer is correct; the experiment presented in Figure 5A did not demonstrate a statistically significant survival difference between these two groups. There may be a trend to improved survival in the combination group but since survival was impressive following both treatments, many more mice would be required to detect a statistically significant difference.

We also note that the two deaths in combination-treated mice both occurred very early in the experiment, indicating that they may not have died due to treatment failure. Moreover, after these two early deaths, the other 21 mice all survived until experimental endpoint 4 months later, demonstrating durable tumour control. One of the two mice that died early in this group was culled because of weight loss, rather than the weight gain due to tumour growth and production of ascites that is more typical of this model. At necropsy this mouse was found to have obstruction of the stomach and a cluster of adjacent tumour nodules were identified. It therefore appears that this mouse died due to the specific anatomical location of the engrafted tumour cells. The other mouse had a very high luminescence at the upper limit of our predefined criteria for enrolment in the study. We excluded mice above this level because we considered that their intraperitoneal tumours were too widely spread to be effectively controlled. Thus it is possible that these two mice in the combination group, are in fact experimental outliers that could have skewed our results. This could only be resolved by a larger study enrolling a greater number of mice. We agree with this reviewer that it will be informative to include a discussion of this more nuanced interpretation of this experiment and we have added the following to clarify:

Page 17, Line 423-434:

***dI922-947* alone (n=22) and the combination of *dI922-947* and carvedilol (n=23) both significantly extended survival compared to untreated control mice (Fig.5A, $P=0.0001$). Mice in both groups had extremely favourable outcomes with 71.4% and 88.9% of mice respectively surviving for the duration of the experiment thus we did not detect a statistically significant survival difference between these two very effective treatments. Interestingly, only two mice died following combination treatment and both of these deaths occurred very early in the experiment. After these two early deaths, the other 21 mice all survived until experimental endpoint 4 months later. One of these mice was culled because of weight loss, rather than the weight gain due to tumour growth and production of ascites that is more typical of this model. At necropsy this mouse was found to have gastric obstruction due to a small cluster of tumour nodules adjacent to the stomach. The other mouse had a very high baseline luminescence, which was at the upper limit of our predefined criteria for enrolment in the study consistent with extensive disease even at this very early time point.**

*'...Moreover, at weeks 10, the relative tumor radiance of the combination therapy group was not significantly different from that of the *dI922-947* monotherapy group (Figure 5E). Please comment.'*

Mice were enrolled once tumour radiance had reached our predefined baseline of 10^5 photons per second per cm^2 . Radiance increased beyond this following enrolment in all four treatment groups. Treatment with both *dI922-947* alone and with combination therapy was able to reduce tumour radiance to below this baseline level and it is true that there was no significant difference in tumour radiance at week ten. The important finding in this experiment is that this reduction to baseline radiance was achieved much earlier in combination-treated mice occurring two weeks after completion of treatment, whereas this took four weeks longer to achieve when mice were treated with *dI922-947* alone. We have emphasized this in the manuscript:

Page 17, Line 441-446:

In contrast, treatment with virus-alone and with the combination of virus and carvedilol effectively controlled tumour growth such that radiance reduced below the baseline level required for initial randomisation in the experiment ($10^5 - 10^7$ p/s/ cm^2/sr). Importantly, in combination-treated mice, this reduction to baseline was achieved only 2 weeks following completion of treatment. Tumour control was slower following single agent *dI922-947* and the same reduction in radiance took four weeks longer to achieve (Fig.5E).

We have also addressed both of these points in the discussion:

Page 20, Line 513-521:

Our most impactful finding however, was that the combination of carvedilol and oncolytic adenoviruses had impressive anti-cancer activity even in platinum-resistant intraperitoneal models. Combination therapy was able to rapidly eliminate tumour-associated bioluminescence only 2 weeks following completion of treatment. Single agent *dI922-947* did eventually achieve tumour control but this reduction to baseline radiance took four weeks longer than the combination of *dI922-947* and carvedilol. Thus although combination treatment controlled tumours more rapidly, both the combination and *dI922-947* alone were ultimately highly effective. This was reflected in the impressive survival benefit seen with both treatments. A much greater number of mice would be required to detect a statistical difference between the two groups.

2. *'The study showed carvedilol co-treatment was associated with induction of innate immunity. Relevant study data were obtained from immunodeficient xenograft models. Nude mice lack T cells. A large number of studies have confirmed that T cells play an important role in oncolytic virus mediated antitumor effects. Therefore, the synergistic effects and mechanisms of combination therapy should be further verified in an immunocompetent murine model.'*

Adenoviruses are species-specific and so human adenoviruses cannot replicate in non-human cells. The oncolytic adenovirus, *d1922-947*, is not armed and so its anti-tumour effect is reliant on intratumoural replication. The resulting amplification of viral dose is advantageous, particularly in the treatment of disseminated malignancies like high grade serous cancer, which usually metastasises widely within the peritoneal cavity. Here we used intraperitoneal human ovarian cancer xenografts in nude mice to model this clinical situation, although we do agree with this reviewer that the absence of T cells is an important limitation of our study.

Immunocompetent murine models have been used to investigate T-cell immunity but only following direct intratumoural injection of non-replicating adenoviruses into localised tumours. Adenoviral species-specificity means that it has not yet been possible to evaluate the role of T cells in the antitumour response to *replicating* oncolytic adenoviruses, particularly following systemic delivery to mice with disseminated cancer. The recent development of humanized mice in which immunodeficient mice are engrafted with peripheral blood mononuclear cells or haematopoietic stem cells, may in future offer a solution to this pervasive challenge. Potentially, an ovarian cancer intraperitoneal xenograft could be generated in such a mouse but we are not aware of any published or commercial evidence that such an ovarian cancer model has been created to date.

We strongly believe that developing such a model would be very time consuming, costly and high risk. So while we agree that it would be fascinating to explore the combination of carvedilol and *d1922-947* in an immunocompetent murine system, we do not think that it is practically possible to pursue such a strategy for the current manuscript.

We agree that it is important to discuss this limitation of our study and the potential future avenues to achieve this and we have added the following to our discussion section:

Page 22, Line 554-572:

T cells are known to play a key role in the anti-tumour efficacy of oncolytic viral agents ¹ and this has been elegantly described following T-VEC ². An important limitation of our study is that we were unable to interrogate this aspect of oncolytic activity in the immunodeficient xenograft models used here. Adenoviruses are species-specific, so human adenoviruses cannot replicate in non-human cells and since *d1922-947* is not armed, its anti-tumour effect is entirely reliant on its ability to specifically replicate within human tumours ³. The resulting amplification of viral dose is advantageous in the treatment of disseminated malignancies like the clinically representative intraperitoneal models of HGSC we used here. T-cell immunity induced by oncolytic adenoviruses has been modelled in immunocompetent tumour-bearing mice but only following direct intratumoural injection of non-replicating adenoviruses into localised tumours ^{1,4,5}. Adenoviral species-specificity means that it has not yet been possible to evaluate the role of T cells in the antitumour response to replicating oncolytic adenoviruses, particularly following systemic delivery to mice with disseminated cancer. The recent development of humanized mice in which immunodeficient mice are engrafted with peripheral blood mononuclear cells or haematopoietic stem cells (reviewed in ^{6,7}), may in future offer a solution to this pervasive challenge. Potentially,

an ovarian cancer intraperitoneal xenograft could be created in such a mouse but we are not aware of any published or commercial evidence that such an ovarian cancer model has been generated to date. Investigating the combination of carvedilol and oncolytic *dI922-947* in an immunocompetent setting is an obvious next step and since carvedilol is a comparatively safe drug, this could realistically include clinical trials.

*3. 'Mice were administered by IP injection of *dI922-947* (5×10^7 vp in 400 μ l PBS) and by oral gavage of carvedilol (10mg/kg in PBS +1% methylcellulose). Treatments were administered once daily for 5 days. What is the basis for the therapeutic schedule?'*

The administration schedule of *dI922-947* is established in our group and was based on our original work investigating *dI922-947* in a different intraperitoneal ovarian cancer model ^{8,9}. Here we used a lower viral dose of 5×10^7 vp in 400 μ l PBS so that we could detect an incremental benefit with the addition of carvedilol.

The carvedilol dose is based on published work using between 5 and 15mg/kg ^{5,10,11} and since it is an oral drug, we administered carvedilol to mice by oral gavage. Our animal licence restricts the administrations of substances to 21 per mouse regardless of the route used. This figure includes initial tumour cell injection and administration of Firefly luciferase for bioluminescence imaging. Since each mouse was imaged 10 times during the course of the experiment, we were limited to five doses of carvedilol. Finally, we elected to administer these five doses of carvedilol on the same day as *dI922-947* to reflect the usual management of human patients, in which combination therapies are most commonly given on the same treatment day. We have explained this in the methods section:

Page 10, Line 243-254:

Mice were randomly allocated to treatment groups and treatments were administered by IP injection of *dI922-947* (5×10^7 vp in 400 μ l PBS) and by oral gavage of carvedilol (10mg/kg in PBS +1% methylcellulose). Treatments were administered once daily for 5 days on day 21-26 and in all cases, vehicle controls were administered by the same route. The administration schedule of daily treatment for five days is based on our previously published work in different murine models ^{8,9} although this time we used a lower dose of 5×10^7 vp to detect an incremental benefit with the addition of carvedilol. The carvedilol dose is based on publications using between 5 and 15mg/kg daily ^{5,10,11} and since it is an oral drug, we administered carvedilol to mice by oral gavage. Our animal license restricted us to five carvedilol treatments and so we elected to administer carvedilol on the same days as *dI922-947* to reflect the usual management of human patients, in which combination therapies are most commonly given on the same treatment day.

Minor:

*1. In the title of the legend of Figure 1, "*dI922-941*" should be changed to "*dI922-947*".*

We have made this change.

*2. In Figure 6F, the relative tumor radiance of combination therapy group and *dI922-947* monotherapy group should be statistically analyzed.*

We presume this reviewer is in fact referring to Figure 5F. We have annotated Figure 5F accordingly.

Reviewer #2 (Remarks to the Author):

We are delighted that this reviewer '*support(s) the publication of this study*' and did not identify any major issues. Our replies to their minor comments are below:

Minor :

'On page 19 lines 397-399, the author state: "Despite the expected anti-inflammatory effect of carvedilol, these data therefore demonstrate that the combination of dI922-947 and carvedilol paradoxically augmented virus-induced inflammatory signalling in platinum-resistant HGSC cells in vitro." Could the anti-inflammatory effect of carvedilol favour viral replication? I wonder whether the sequential administration (rather than concomitant use of carvedilol/virus) could help address this issue. The authors might consider discussing this possibility.'

The experiments shown in Figure 4 we used the same workflow as the initial drug screen with administration of dI922-947 24 hours after cells were seeded followed by carvedilol at 48 and 120 hours. We have clarified this in the results section

Page 16, Line 401-403:

To investigate this hypothesis, Ov4Carbo cells were treated with dI922-947 (MOI10), carvedilol (10µM) or the combination using the same workflow as the initial drug screen (dI922-947 or control at 24 hours, followed by carvedilol or control at 48 and 120 hours). Proteins were harvested over time.

We agree that the anti-inflammatory effect of carvedilol could favour viral replication and that the sequencing of virus and carvedilol treatment could be relevant. We have highlighted this in the discussion:

Page 21, Line 534-543:

Our data show that *in vitro* treatment with dI922-947, followed by targeting of β-arrestins by carvedilol, was able to induce downstream activity of Akt and NFκB. *In vivo*, simultaneous treatment with dI922-947 and carvedilol induced a local inflammatory cell infiltration together with systemic release of inflammatory and anti-viral cytokines including IFNγ, TNFα and IL-27/28. This inflammatory response could represent either a cause or a consequence of increased oncolytic activity and interestingly our findings appear to contrast with current literature suggesting that carvedilol is associated with reduced inflammatory markers in disease models such as diabetes and acute pancreatitis¹². A possible explanation is that the known anti-inflammatory effect of carvedilol promoted viral replication. Experiments comparing different scheduling of the two agents could be used to further explore this possibility.

'Supplementary tables 1 & 2: please clarify why some plates/ plate positions are marked in red and provide a color legend'

The red text for the plate positions is an error. We have now corrected this and all the text in the first two columns (entitled 'Plate' and 'Well') is now black.

We applied a colour code to the z-scores shown in this tables and we have now added an explanation of this in the legends for Supplementary Tables 1 and 2.

Reviewer #3 (Remarks to the Author):

'The authors declare that observed a survival advantage following single agent carvedilol treatment (lines 481-482), has the anticancer activity of carvedilol alone been already described? If so, please discuss and add appropriate references.'

To our knowledge the specific anticancer activity of carvedilol has not previously been described. There is however epidemiological evidence of a survival benefit in cancer patients taking β -blockers for other indications. These studies do not provide detail on the specific effect of carvedilol. We have clarified this in the discussion:

Page 20, Line 500-508:

Interestingly, improved ovarian cancer-specific survival has previously been demonstrated in epidemiological studies of patients receiving non-specific β -blockers for other indications. In one study, 344 HGSC patients receiving a range of β -blockers had a better overall survival of 90 compared to 38.2 months¹³. In another study of ovarian cancer patients over the age of 60, the use of non-selective β -blockers (151 patients) was again associated with better survival (Hazard Ratio =0.579;¹⁴. It has been postulated that this could be due to the known anti-proliferative effects of these drugs¹⁵⁻¹⁷ but both studies group multiple different β -blocking drugs in their analysis and provide no detail on the specific effect of carvedilol. Moreover, we are not aware of any studies describing the anti-cancer activity of carvedilol in human patients.

'The authors describe the mechanism of action of the combination treatment that is mediated by β -arrestins, however I would discuss how the anticancer activity of adenovirus might be elicited, i.e known or recently reported cell death mechanisms reported in literature that might account for cancer cell survival decrease or recall of immune cells (apoptosis, immunogenic cell death, etc).'

We have agree with this important comment and have added the following to the discussion section:

Page 22, Line 545-552:

Oncolytic viruses hold promise as a novel means to overcome immune evasion in cancers and have been modified to elicit a favourable anti-tumour immune response in otherwise immune-silent cancer cells¹⁸⁻²⁰. This is predominantly achieved through the release and presentation of antigens following virus-induced cell death. Adenovirus-induced cell death however, is a multifactorial process including autophagy, apoptosis, necrosis and pyroptosis in addition to induction of an immune response²¹⁻²⁵. Further studies to determine the mechanism of cell death following the combination of carvedilol and dI922-947 would likely help to unravel the relationship between the improved viral oncolysis and the altered inflammatory response that we observed.

'Finally I think that the discussion should be improved.'

We have re-written our discussion and trust that our extensive additions and modifications at lines **500-508, 513-521, 534-543, 545-552 and 554-572** sufficiently address this reviewer's concern.

References

1. Teijeira Crespo A, Burnell S, Capitani L, et al: Pouring petrol on the flames: Using oncolytic virotherapies to enhance tumour immunogenicity. *Immunology* 163:389-398, 2021
2. Ramelyte E, Tastanova A, Balazs Z, et al: Oncolytic virotherapy-mediated anti-tumor response: a single-cell perspective. *Cancer Cell* 39:394-406 e4, 2021
3. Heise C, Hermiston T, Johnson L, et al: An adenovirus E1A mutant that demonstrates potent and selective systemic anti-tumoral efficacy. *Nat Med* 6:1134-9, 2000
4. Cervera-Carrascon V, Quixabeira DCA, Santos JM, et al: Adenovirus Armed With TNFa and IL2 Added to aPD-1 Regimen Mediates Antitumor Efficacy in Tumors Refractory to aPD-1. *Front Immunol* 12:706517, 2021
5. Chen YL, Chung SY, Chai HT, et al: Early Administration of Carvedilol Protected against Doxorubicin-Induced Cardiomyopathy. *J Pharmacol Exp Ther* 355:516-27, 2015
6. Tian H, Lyu Y, Yang YG, et al: Humanized Rodent Models for Cancer Research. *Front Oncol* 10:1696, 2020
7. Yin L, Wang XJ, Chen DX, et al: Humanized mouse model: a review on preclinical applications for cancer immunotherapy. *Am J Cancer Res* 10:4568-4584, 2020
8. Lockley M, Fernandez M, Wang Y, et al: Activity of the adenoviral E1A deletion mutant dl922-947 in ovarian cancer: comparison with E1A wild-type viruses, bioluminescence monitoring, and intraperitoneal delivery in icodextrin. *Cancer Res* 66:989-98, 2006
9. Browne A, Tookman LA, Ingemarsdotter CK, et al: Pharmacological Inhibition of beta3 Integrin Reduces the Inflammatory Toxicities Caused by Oncolytic Adenovirus without Compromising Anticancer Activity. *Cancer Res* 75:2811-21, 2015
10. Shimada K, Hirano E, Kimura T, et al: Carvedilol reduces the severity of atherosclerosis in apolipoprotein E-deficient mice via reducing superoxide production. *Exp Biol Med (Maywood)* 237:1039-44, 2012
11. Wang D, Chen Y, Jiang J, et al: Carvedilol has stronger anti-inflammation and anti-virus effects than metoprolol in murine model with coxsackievirus B3-induced viral myocarditis. *Gene* 547:195-201, 2014
12. Amirshahrokhi K, Zohouri A: Carvedilol prevents pancreatic beta-cell damage and the development of type 1 diabetes in mice by the inhibition of proinflammatory cytokines, NF-kappaB, COX-2, iNOS and oxidative stress. *Cytokine* 138:155394, 2021
13. Watkins JL, Thaker PH, Nick AM, et al: Clinical impact of selective and nonselective beta-blockers on survival in patients with ovarian cancer. *Cancer* 121:3444-51, 2015
14. Baek MH, Kim DY, Kim SO, et al: Impact of beta blockers on survival outcomes in ovarian cancer: a nationwide population-based cohort study. *J Gynecol Oncol* 29:e82, 2018
15. Fujio H, Nakamura K, Matsubara H, et al: Carvedilol inhibits proliferation of cultured pulmonary artery smooth muscle cells of patients with idiopathic pulmonary arterial hypertension. *J Cardiovasc Pharmacol* 47:250-5, 2006
16. Erguven M, Yazihan N, Aktas E, et al: Carvedilol in glioma treatment alone and with imatinib in vitro. *Int J Oncol* 36:857-66, 2010
17. Cleveland KH, Liang S, Chang A, et al: Carvedilol inhibits EGF-mediated JB6 P+ colony formation through a mechanism independent of adrenoceptors. *PLoS One* 14:e0217038, 2019
18. McGray AJR, Huang RY, Battaglia S, et al: Oncolytic Maraba virus armed with tumor antigen boosts vaccine priming and reveals diverse therapeutic response patterns when combined with checkpoint blockade in ovarian cancer. *J Immunother Cancer* 7:189, 2019
19. Murphy JP, Kim Y, Clements DR, et al: Therapy-Induced MHC I Ligands Shape Neo-Antitumor CD8 T Cell Responses during Oncolytic Virus-Based Cancer Immunotherapy. *J Proteome Res* 18:2666-2675, 2019
20. Thomas ED, Meza-Perez S, Bevis KS, et al: IL-12 Expressing oncolytic herpes simplex virus promotes anti-tumor activity and immunologic control of metastatic ovarian cancer in mice. *J Ovarian Res* 9:70, 2016
21. Braithwaite AW, Russell IA: Induction of cell death by adenoviruses. *Apoptosis* 6:359-70, 2001

22. Tazawa H, Kagawa S, Fujiwara T: Oncolytic adenovirus-induced autophagy: tumor-suppressive effect and molecular basis. *Acta Med Okayama* 67:333-42, 2013
23. Baird SK, Aerts JL, Eddaoudi A, et al: Oncolytic adenoviral mutants induce a novel mode of programmed cell death in ovarian cancer. *Oncogene* 27:3081-90, 2008
24. Tazawa H, Kuroda S, Hasei J, et al: Impact of Autophagy in Oncolytic Adenoviral Therapy for Cancer. *Int J Mol Sci* 18, 2017
25. Radke JR, Siddiqui ZK, Figueroa I, et al: E1A enhances cellular sensitivity to DNA-damage-induced apoptosis through PIDD-dependent caspase-2 activation. *Cell Death Discov* 2:16076, 2016

REVIEWERS' COMMENTS:

Reviewer #1 (Remarks to the Author):

I have some other comments:

1. The authors declare that adenoviruses are species-specific, so human adenoviruses cannot replicate in non-human cells (Page 23, line 569) and adenoviral species-specificity means that it has not yet been possible to evaluate the role of T cells in the antitumor response to replicating oncolytic adenoviruses, particularly following systemic delivery to mice with disseminated cancer (Page 23, line 575-577). These statements are incorrect. Many studies have reported that oncolytic adenoviruses could replicate in non-human tumors and inhibited tumor growth in immunocompetent mice [1-3]. Furthermore, T cells play an important role in the antitumor effect of oncolytic adenovirus [3].

[1] G. Hallden, R. Hill, Y. Wang, A. Anand, T.C. Liu, N.R. Lemoine, J. Francis, L. Hawkins, D. Kim, Novel immunocompetent murine tumor models for the assessment of replication-competent oncolytic adenovirus efficacy, *Mol Ther* 8(3) (2003) 412-24.

[2] H. Huang, Y. Liu, W. Liao, Y. Cao, Q. Liu, Y. Guo, Y. Lu, Z. Xie, Oncolytic adenovirus programmed by synthetic gene circuit for cancer immunotherapy, *Nat Commun* 10(1) (2019) 4801.

[3] G. Shi, Q. Yang, Y. Zhang, Q. Jiang, Y. Lin, S. Yang, H. Wang, L. Cheng, X. Zhang, Y. Li, Q. Wang, Y. Liu, Q. Wang, H. Zhang, X. Su, L. Dai, L. Liu, S. Zhang, J. Li, Z. Li, Y. Yang, D. Yu, Y. Wei, H. Deng, Modulating the Tumor Microenvironment via Oncolytic Viruses and CSF-1R Inhibition Synergistically Enhances Anti-PD-1 Immunotherapy, *Mol Ther* 27(1) (2019) 244-260.

Reviewer #2 (Remarks to the Author):

The authors satisfactorily addressed the reviewer concerns.

Reviewer #3 (Remarks to the Author):

I have no further comments on that paper, the authors addressed all my concerns!

REVIEWERS' COMMENTS:

Reviewer #1 (Remarks to the Author):

I have some other comments:

1. The authors declare that adenoviruses are species-specific, so human adenoviruses cannot replicate in non-human cells (Page 23, line 569) and adenoviral species-specificity means that it has not yet been possible to evaluate the role of T cells in the antitumor response to replicating oncolytic adenoviruses, particularly following systemic delivery to mice with disseminated cancer (Page 23, line 575-577). These statements are incorrect. Many studies have reported that oncolytic adenoviruses could replicate in non-human tumors and inhibited tumor growth in immunocompetent mice [1-3]. Furthermore, T cells play an important role in the antitumor effect of oncolytic adenovirus [3].

[1] G. Hallden, R. Hill, Y. Wang, A. Anand, T.C. Liu, N.R. Lemoine, J. Francis, L. Hawkins, D. Kirn, Novel immunocompetent murine tumor models for the assessment of replication-competent oncolytic adenovirus efficacy, *Mol Ther* 8(3) (2003) 412-24.

[2] H. Huang, Y. Liu, W. Liao, Y. Cao, Q. Liu, Y. Guo, Y. Lu, Z. Xie, Oncolytic adenovirus programmed by synthetic gene circuit for cancer immunotherapy, *Nat Commun* 10(1) (2019) 4801.

[3] G. Shi, Q. Yang, Y. Zhang, Q. Jiang, Y. Lin, S. Yang, H. Wang, L. Cheng, X. Zhang, Y. Li, Q. Wang, Y. Liu, Q. Wang, H. Zhang, X. Su, L. Dai, L. Liu, S. Zhang, J. Li, Z. Li, Y. Yang, D. Yu, Y. Wei, H. Deng, Modulating the Tumor Microenvironment via Oncolytic Viruses and CSF-1R Inhibition Synergistically Enhances Anti-PD-1 Immunotherapy, *Mol Ther* 27(1) (2019) 244-260.

We now acknowledge these papers in our discussion section at lines 389-393 :

'An important limitation of our study is that we were unable to interrogate this aspect of oncolytic activity in the immunodeficient xenograft models used here. Adenoviruses are species-specific, so human adenoviruses replicate poorly in non-human cells although it has been possible to achieve modest intratumoural replication in specific murine cell lines⁵², following viral modification⁵³ and certain combination therapies⁵⁴.'